# Perceived Feasibility of Endoscopic Ultrasound-Guided Gastroenteric Anastomosis: An Italian Survey

**DOI:** 10.3390/medicina58040532

**Published:** 2022-04-12

**Authors:** Ilaria Tarantino, Emanuele Sinagra, Cecilia Binda, Alessandro Fugazza, Arnaldo Amato, Marcello Maida, Andrea Lisotti, Stefano Francesco Crinò, Giovanni Aragona, Carlo Fabbri, Andrea Anderloni

**Affiliations:** 1Department of Diagnostic and Therapeutic Services, Endoscopy Service, IRCCS-ISMETT, 90100 Palermo, Italy; itarantino74@gmail.com; 2Gastroenterology and Endoscopy Unit, Fondazione Istituto G. Giglio, Contrada Pietrapollastra Pisciotto, Cefalù, 90015 Palermo, Italy; 3Gastroenterology and Digestive Endoscopy Unit, Forlì-Cesena Hospitals, AUSL Romagna, 47121 Forli, Italy; cecilia.binda@gmail.com (C.B.); carlo.fabbri@auslromagna.it (C.F.); 4Humanitas Clinical and Research Center IRCCS, Gastroenterology & Endoscopy Department, 20089 Rozzano, Italy; alessandro.fugazza@humanitas.it; 5Division of Digestive Endoscopy and Gastroenterology, Valduce Hospital, 22100 Como, Italy; arnamato@gmail.com; 6Gastroenterology and Endoscopy Unit, S. Elia-Raimondi Hospital, 93100 Caltanissetta, Italy; marcello.maida@hotmail.it; 7Gastroenterology Unit, Hospital of Imola, University of Bologna, 40026 Imola, Italy; lisotti.andrea@gmail.com; 8Gastroenterology and Digestive Endoscopy Unit, The Pancreas Institute, G.B. Rossi University Hospital, 37134 Verona, Italy; stefanocrino@hotmail.com; 9Gastroenterology and Hepatology Unit, Ospedale Civile, Sitodell’azienda AUSL di Piacenza, 29121 Piacenza, Italy; g.aragona@ausl.pc.it; 10Unità Operativa Complessa di Gastroenterologia ed Endoscopia Digestiva, Fondazione IRCCS Policlinico San Matteo, 27100 Pavia, Italy; *andrea_anderloni@hotmail.com*

**Keywords:** EUS, survey, endoscopic ultrasound-guided gastroenteric anastomosis, safety, feasibility

## Abstract

*Background and Objectives*: Endoscopic ultrasound-guided gastroenteric anastomosis (EUS-GEA) using lumen-apposing metal stents (LAMS) is emerging as a minimally invasive alternative to surgery across several indications. The aim of this survey is to investigate the perceived feasibility of this technique nationwide, within a working group skilled in interventional endosonography. *Materials and Methods:* Endoscopists were asked to answer to 49 items on a web-based questionnaire about expertise, peri- and intra-procedural aspects in the three main settings of EUS-GEA performance, budget/refund, and future perspectives. Statistical analysis was performed through SPSS^®^ (IBM Corp. Released 2017. IBM SPSS Statistics for Windows, Version 25.0. Armonk, NY: IBM Corp.). *Results*: Sixty endosonographers belonging to forty Italian centers were I-EUS app users and were all invited to participate. In total, 29 participants from 24 Italian centers completed the survey. All the participants were endosonographers with a broad range of experience both in the field of EUS (only 10.3% with more than 20 years of experience) and duodenal stenting (only 6.9% placed more than 10 stents in 2020), whereas 86.2% also performed ERCP. A total of 27.6% of participants performed EUS-GEA (3.4% more than 20 during their career); on the other hand, 79.3% of participants routinely performed drainage of peri-pancreatic fluid collections, 62.1% performed biliary drainage, and 62.1% performed gallbladder drainage with LAMS. A total of 89.7% of participants thought that EUS-GEA could be useful in their daily clinical practice, with 100% concluding that this procedure will need to be performed in referral centers in the near future; however, in 55.2% of cases, organizational obstacles may occur and affect the diffusion of the procedure. With regard to indications: 44.8% of participants performed the procedure with palliative intent for malignant indication (96.6% pancreatic adenocarcinoma), and 13.6% also for benign indication. A total of 20.7% of participants experienced adverse events (none severe or fatal, 66.6% moderate). A total of 62.1% of participants considered the procedure technically challenging, although 82.8% considered the risk of adverse events acceptable when considering the benefit. *Conclusions*: To our knowledge, this is the first survey assessing the perceived feasibility of EUS-guided anastomoses after its advent. There are currently wide variations in practice nationwide, which demonstrate a need to define technical, qualitative, and peri-procedural requirements to carry out this procedure. Therefore, a standardization of these requirements is needed in order to overcome the technical, economical, and organizational obstacles relative to its diffusion.

## 1. Introduction

Significant advancements in interventional EUS have been recently described [1,2,3,4,5,6,7].

Moreover, endoscopic ultrasound-guided gastrointestinal anastomosis (EUS-GEA) with lumen-apposing metal stents (LAMS) is a new and emerging technique that has gained increasing interest in recent years [8].

LAMS is a recently developed device, with a “barbell” shape and flanged ends that provide anti-migration properties. This stent was originally created for pancreatic fluid collection drainage because of its inner lumen diameter, which is significantly wider than a plastic or traditional self-expanding metal stent (SEMS), and allows the entrance of an endoscope for collections and the performance of direct endoscopic necrosectomy (DEN). Over time, many indications have been proposed, including the EUS-guided drainage of pancreatic fluid collections, biliary drainage, pancreatic duct drainage, gallbladder drainage, and gastroenterostomy [9]. In this last setting, in the first animal study performed by Binmoeller and Shah, a gastrojejunostomy was created under EUS guidance in five animals (four survival). The stents remained fully patent in all animals throughout the implantation period (up to 4.5 weeks) and were easily removed [10].

Many studies have been recently published showing that these minimally invasive procedures provide great opportunities to avoid invasive surgical procedures [2].

Examples of the potential clinical applications include bypassing malignant and benign gastric outlet obstruction, providing access to the pancreatic–biliary tree in those who have undergone Roux-en-Y gastric bypass, and relieving pancreatic–biliary symptoms in afferent limb syndrome [2]. Although these new devices facilitate the stent deployment, mainly thanks the electrocautery-enhanced tip that allows the procedure to be performed in a single step, procedure-related adverse events are not negligible, being in some cases severe and thus requiring maximal expertise in interventional endoscopy for their management [5,11]; moreover, candidates for this procedure are usually hard-to-treat patients in life-threatening conditions, requiring multidisciplinary approaches. Nevertheless, no guidelines or consensus about training and peri-procedural aspects of LAMS placement exist, so it is currently difficult to define the standard of care for the use of this device, especially in the setting of EUS-GEA, which still represents an off-label indication.

Indeed, there is still little knowledge among the professional figures involved in this procedure, as the management paths are generally entrusted to the individual professional or single center, and a standardized view of the procedure itself is still lacking.

The aim of this survey is to investigate the perceived feasibility of this procedure nationwide, with the purpose of being helpful in understanding practice patterns and identifying areas of controversy to guide future research towards a beneficial standardization.

## 2. Materials and Methods

The rationale for developing the present study was to focus on the perceived feasibility and safety of EUS-GEA in all its indications, both in oncological settings and in benign diseases. The survey was developed according to CHERRIES guidelines (see Appendix A) [12].

A web-based 49-question survey on EUS-GEA was submitted during a temporal trend of three months (November 2019–January 2020), among a working group of endoscopists skilled (at least 100 EUS performed individually) in interventional EUS and LAMS placement, from 40 Italian centers. In detail, by using a smart phone app called I-EUS, designed for all the physicians involved in interventional EUS, participants entered their personal experience about the perceived feasibility and safety of EUS-GEA. The questionnaire was reviewed by three experts (C.F., A.A., and I.T.) (with at least an experience of 20 years of practice in the setting of EUS) from three different Italian referral centers. For some questions, there was the possibility to choose between just one of the answers, while for others, participants were allowed to select multiple options. Data were reported anonymously on a database (Excel, Open Office). Consent to the use of data for research purposes only was implicit in the choice to join the survey.

### 2.1. Design of the Questionnaire

The questions were grouped under several sections:

EXPERTISE AS LAMS USER: background demographics, availability of radiology room, and experience in advanced endoscopy (EUS, ERCP, and duodenal stenting) and in LAMS positioning.

PERCEIVED FEASIBILITY OF THE TECHNIQUE: utility of EUS-GEA in a previous clinical situation in their experience, potential diffusion of the procedure nationwide and eventual obstacles to its diffusion, willingness to deepen the knowledge about the procedure, and experience about EUS-GEA performance and its indications (with focus in the malignant setting).

PERI-PROCEDURAL MANAGEMENT: adverse events and their severity, reasons for disagreeing to carry out the procedure with the other specialists, satisfaction within the multidisciplinary team (with a focus on the role of the surgeons) regarding the availability of the technique in their own center, and reasons for eventual satisfaction.

PROCEDURAL ASPECTS: technical considerations about the procedure and the setting and con division with referral centers and training.

EUS-GEA or EUS-JJ (jejeunum-jejeunum-anastomosis) IN TREATING AFFERENT LIMB SYNDROME: previous experience, utility of procedure in previously experienced clinical situations, willingness to deepen knowledge about the procedure, satisfaction in the multidisciplinary team (with a focus on the role of the surgeons) about the availability of the technique in their own center, reasons for eventual satisfaction, eventual obstacles to its diffusion, and the utility of the con division with referral centers.

EUS-GG (gastro-gastrostomy) IN TREATING PATIENTS WITH ALTERED GASTROINTESTINAL ANATOMY OR WITH PREVIOUS BARIATRIC SURGERY: previous experience, utility of EUS-GG in previously experienced clinical situations, willingness to deepen knowledge about the procedure, satisfaction in the multidisciplinary team (with a focus on the role of the surgeons) about the availability of the technique in their own center, reasons for eventual satisfaction, eventual obstacles to its diffusion, and the utility of the con division with referral centers.

### 2.2. Statistical Analysis

Categorical variables were summarized with frequencies and proportions. Statistical analysis was performed through SPSS^®^ (IBM Corp. Released 2017. IBM SPSS Statistics for Windows, Version 25.0. Armonk, NY, USA: IBM Corp.).

## 3. Results

A total of 60 endosonographers belonging to 40 Italian centers were I-EUS app users, and all were invited to participate. A total of 29 participants from 24 centers completed the survey. Only questions answered by at least 50% of the participants were reported in the results.

### 3.1. SectionI: Expertise Lams Users

The first section of the questionnaire was composed of 12 questions.

The experience of participants in the EUS setting had extremely variable results: 41.4% (12/29) had less than 5 years of experience, whereas only 6/29 had more than 15 years of experience (Figure 1). A total of 25 out of 29 participants were also involved in carrying out ERCP. With regard to duodenal stent placement, only 6.9% at the center placed more than 10 stents in 2020 (2/29), and 18/29 placed at least 6–10 duodenal stents in the same period (Figure 2). Interestingly, 3/29 (10.3%) participants did not possess any duodenal stent placement in their endoscopic experience (Figure 3).

Interestingly, 4/29 participants (13.8%) declared that there was no radiology room available in their endoscopy unit (Figure 3).

A total of eight participants actively performed EUS-GEA; of these, only three had performed more than ten procedures in their endoscopic experience, while the others had an overall experience of less than ten EUS-GEA.

In contrast, 23/29 (79.3%) participants had experience in performing drainage through LAMS on peri-pancreatic fluid collections, 18/29 (62.1%) on common bile duct, and 18/29 (62.1%) on gallbladder (Figure 4).

### 3.2. SectionII: Perceived Feasibility of the Technique

The second section of the questionnaire was composed of 11 questions.

A total of eight participants had carried out EUS-GEA in their endoscopic experience, but only three out of seven had performed more than ten procedures (Figure 5). Among the centers involved in performing EUS-GEA (11/29, 37.9%), in seven centers, two endoscopists were dedicated to the procedure, whereas in four other centers, only one endoscopist was dedicated to the procedure.

In total, 26 (89.7%) thought that EUS-GEA could be useful in their daily clinical practice, with 100% concluding that this procedure will need to be performed in referral centers in the near future. However, in 55.2% (16/29) of cases, organizational obstacles may occur and affect the diffusion of the procedure (Figure 5).

Only 2 of 29 participants (6.8%) would not be interested in deepening their knowledge of the technique through dedicated training courses.

With regard to indications, focusing on the palliation of malignant gastric outlet obstruction (GOO): the perceived rate of application of EUS-GEA in the palliative setting was 100% in 13 out of 29 participants (44.8%) (Figure 6). The most frequent (96.6%) malignancy causing GOO was pancreatic adenocarcinoma, whereas the remaining 3.4% was gastric tumor (mainly antral).

However, 3 of 29 participants (10.3%) also reported EUS-GEA for benign indication (two for inflammatory pyloric stenoses and one to perform altered gastro in test in al anatomy).

### 3.3. SectionIII: Peri-Procedural Management

The third section of the questionnaire was composed of four questions.

Out of 15 participants, on surgeon satisfaction regarding the availability of this procedure, only 2 (13.3%) responded negatively. Among the others, 13 surgeons seemed to be in favor of EUS-GEA; this was due to the favorable clinical outcome in 8/13 (61.5%) and organizational issues in the other 5 (38.4%).

Up to six of eight participants experienced adverse events (none severe or fatal, 6/8-75% moderate).

Interestingly, 11 out of 29 participants have managed a patient viable for this approach, but did not execute it due a lack of agreement with other specialists involved in the decision-making process: in 3/11, these specialists were internists or oncologists, and in 8/11, they were surgeons.

### 3.4. SectionIV: Procedural Aspects

The fourth section of the questionnaire was composed of four questions.

A total of 18/29 participants (62.1%) considered the procedure technically challenging (Figure 7). However, despite difficulties, 24/29 participants (82.8%) believed that procedure-related risks are acceptable when compared with potential benefits.

Sharing of the cases, together with referral centers, was supported by all the participants (100%).

### 3.5. SectionV: EUS-GEA or EUS-JJ in Treating Afferent Limb Syndrome

The fifth section of the questionnaire was composed of eight questions.

Only four endoscopists (4/29, 13.7%) performed EUS-GEA to treat afferent limb syndrome. Interestingly, among the remaining 25 participants, only 5 (20%) had experienced at least one case that could be potentially treated with EUS-GEA. Only one participant (3.4%) would not be interested in deepening his knowledge of EUS-GEA in this setting, while three (10.3%) affirmed that the surgeons in their centers would not be interested in the availability of the technique for their patients. However, the main obstacles to the diffusion of the technique in this setting was found to be organizational, technical, and economical in 65.5%, 17.2%, and 10.3% of cases, respectively (Figure 8).

In the 12 centers in which EUS-GEA and EUS-JJ were performed for this indication, only 1 (8.3%) reported the absence of surgeons’ satisfaction about the availability of the procedure, whereas in the other centers, the surgeons were satisfied due to the favorable clinical outcome in 5/11 cases (45.4%), and due to organizational reasons in the remaining 6 cases (54.6%).

Sharing of the cases with referral centers was supported by all the participants (100%).

### 3.6. SectionVI: EUS-GG in Treating Patient with Altered Gastrointestinal Anatomy or with Previous Bariatric Surgery

The sixth section of the questionnaire was composed of 10 questions.

Only 4/29 participants (13.7%) performed EUS-GG in patients with altered anatomy or previous bariatric surgery, with a range of two to five procedures completed.

Interestingly, a total of 15/29 participants (51.7%) reported in their experience only one case susceptible to EUS-GG (Figure 9).

In fact, only 9/29 of the respondents (31%) perform ed enteroscope-assisted ERCP. Only two participants (6.8%) would not be interested in deepening their knowledge on EUS-GG in this setting, while three (10.3%) affirmed that the surgeons they work with would not be interested in the availability of the procedure in their center.

The main barriers to the start-up and dissemination of this technique are supposedly organizational, technical, and economical in 51.7%, 24.1%, and 10.3% of cases, respectively (Figure 10).

The sharing of these cases with referral centers was supported by all the participants (100%).

In the four centers in which EUS-GG is performed for this indication, only one reported that the surgeons were unsatisfied with the availability of the procedure, whereas surgeons were satisfied in the remaining centers because of the favorable clinical outcome (85.7%) and the lower economic burden (14.3%) when compared to surgical treatment.

## 4. Discussion

To our knowledge, this is the first survey assessing the perceived feasibility of EUS-guided anastomoses since its advent.

Several points of interests can be highlighted in this survey.

Firstly, to date, only a small percentage of endosonographers are able to complete this complex procedure, which is still perceived to be technically difficult. Discrepancies in endoscopists’ experiences may also increase the difficulty of the dissemination of the technique. Furthermore, potentially due to its “off-label” indication, the experience of performing EUS-guided anastomoses is still markedly inferior to that of other aforementioned drainage procedures using LAMS.

Secondly, a high percentage of participants consider the procedure to be useful in all the investigated settings (malignant or benign GOO, afferent limb syndrome, and ERCP in altered anatomy), and they reported in their experience the need of EUS-guided anastomoses among their technical equipment. This reflects the almost perfect agreement among participants on their desire to deepen their knowledge on these procedures, and also despite the disagreement of other specialists (i.e., surgeons in a minority of cases) involved in the decision-making process and the organizational, economical, and technical barriers that may limit the diffusion of the technique, especially in non-referral centers.

In the setting of malignant GOO, several randomized trials have compared the outcomes of surgical gastrojejunostomy (SGJ), endoscopic SEMS placement (Figure 11), and EUS-GEA anastomosis (Figure 12) [2,13,14,15,16,17]. The main limitation of SGJ is commonly considered its invasiveness, associated with the occurrence of post-surgical adverse events, as with gastroparesis and post-operative infections [2,13].

The authors of the SUSTENT study, comparing endoscopic SEMS placement and SGJ, despite slow initial symptom improvement, suggested that SGJ was associated with better long-term results and is therefore the treatment of choice for patients with a life expectancy of 2 months or longer. Because stent placement was associated with better short-term outcomes, they considered this treatment preferable for patients expected to live less than 2 months [2,18].

A recent meta-analysis about published data on EUS-GEA including twelve studies (290 patients) showed that malignant GOO was the main procedure indication, with the direct puncture technique the most frequently adopted technique. Pooled technical success and clinical success were 93.5% and 90.1%, respectively, with a pooled total AEs rate of 11.7%, mainly mild/moderate [2,19].

Our experience suggests a higher prevalence of adverse events, even if none were severe or fatal.

With regard to the setting of patients with altered anatomy or previous bariatric surgery, available evidence supports that EUS-GG-guided ERCP may be superior to enteroscopy-assisted ERCP in patients with Roux and Y-Gastric Bypass anatomy in terms of higher technical success and shorter procedural times, offering a similar safety profile [20,21].

Our survey shows that only a small percentage of endoscopistsare able to perform enteroscopy-assisted ERCP. This warrants the interest in EUS-GG-guided ERCP training and in sharing their experiences with referral centers.

In the context of afferent limb syndrome, EUS-JJ seemed to be safe and effective, and, indirect comparison with enteroscopy-assisted procedures, it is suggested that EUS-JJ was associated with a reduced need for re-intervention [22]. Our survey shows that at least 20% of managed patients, according to the participants interviewed, could be treated with EUS-JJ. Furthermore, in this case this warrants the interest in training and in the sharing of their experiences with referral centers.

However, in both these settings, organizational, economical, and technical aspects may also limit the diffusion of the technique.

Despite the increasingly widespread use of LAMS among endoscopy units and the ever-increasing extension of indications in the therapeutic field, the current considerable heterogeneity of this approach, in particular for these “off-label” procedures, seems clear. For this reason, we believed it would have been necessary to take stock of the current situation in order to assess the key points, which can represent a starting point for the standardization of the procedure, and therefore the possible future drafting of guidelines and, above all, to clarify those grey areas which may be the object of future studies.

In our opinion, due to the complexity of these clinical conditions, a multidisciplinary approach in these managed patients is needed, involving oncologists, internists, nutritionists, radiologists, and surgeons, who may represent further key players in this scenario. Such an approach should be guaranteed in order to reach a common agreement on treatment strategies and to be aware about the management of adverse events. Finally, we should consider that many patients who are candidates of EUS-guided procedures are affected by benign diseases in which EUS-guided procedures could be pivotal in their clinical improvement towards recovery [23,24]. This is in contrast with the role of EUS-guided procedures in oncological patients, often with poor life expectancy, who are nevertheless the object of multidisciplinary meetings despite our interventions often being only palliative, without curative intent [25,26].

Indeed, this study presents some limitations.

First, a small sample size was enrolled, even if the performance of EUS-guided anastomoses represents a “niche” of interventional endoscopy, which highlights the fact that the spreading of this procedure is still in its infancy. Furthermore, this feature could explain the low response rate to the survey (almost 50%).

Second, we perceived a discrepancy between the perceived indications of the procedure: in fact, the percentage between palliation in malignancies and benign ranged widely (50–100%) this discrepancy may be due to the difference in endoscopists’ experience and in the geographical allocation of the centers in which the participants were interviewed. Finally, since our study was a survey performed on retrospective data, it was not possible to obtain findings about quality of life and patients’ prognosis.

## 5. Conclusions

In conclusion, there are currently wide variations in practice nationwide, which demonstrate a pressing need to define technical, qualitative, and peri-procedural requirements to carry out these procedures, towards a standardization, in order to overcome the organizational obstacles relative to its diffusion. Therefore, forthcoming studies are needed in order to incorporate EUS-guided anastomoses in the clinical practice.

## Figures and Tables

**Figure 1 medicina-58-00532-f001:**
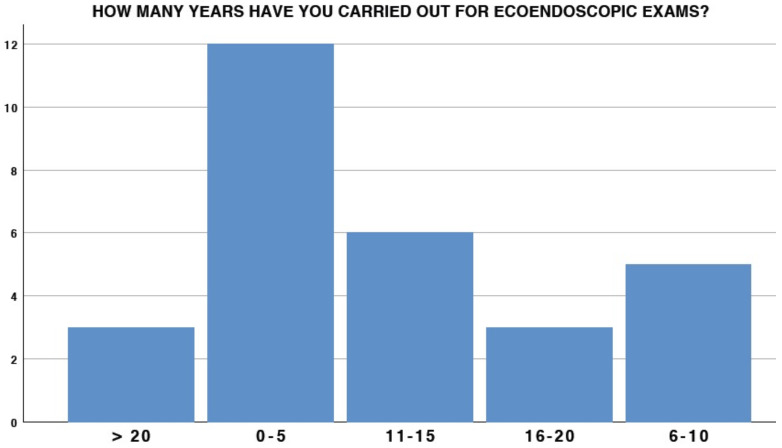
EUS experience.

**Figure 2 medicina-58-00532-f002:**
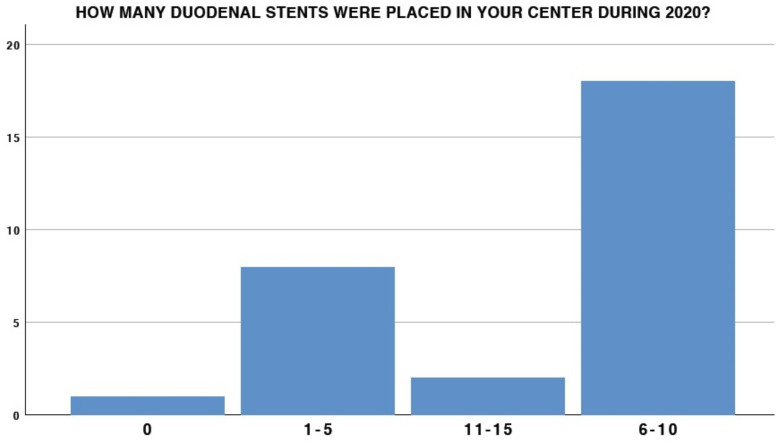
Duodenal stenting experience.

**Figure 3 medicina-58-00532-f003:**
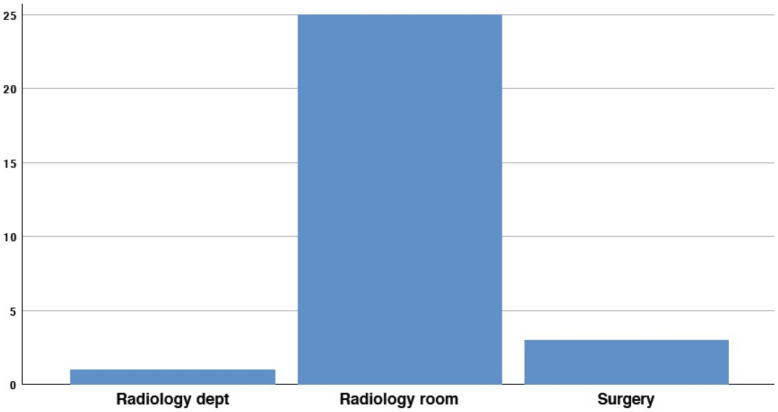
Availability of radiology room.

**Figure 4 medicina-58-00532-f004:**
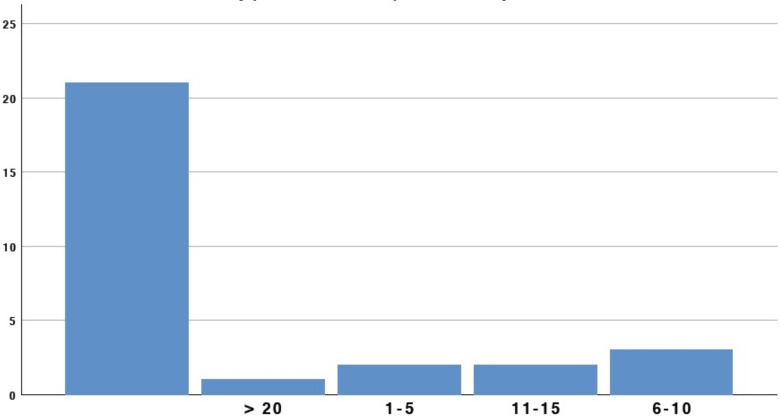
LAMS placement experience.

**Figure 5 medicina-58-00532-f005:**
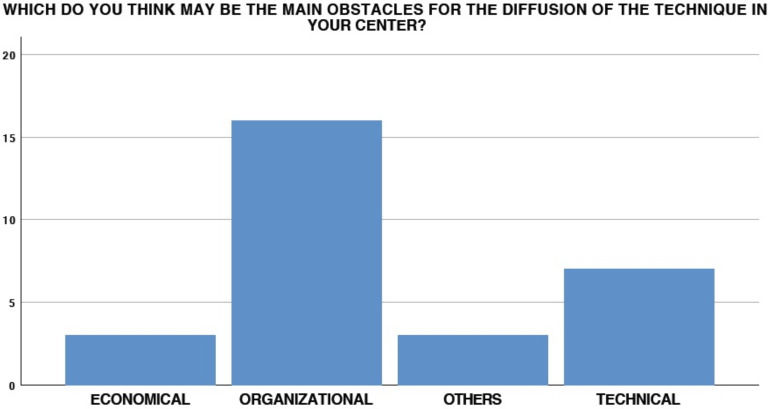
Perceived obstacles in overall EUS-GEA diffusion.

**Figure 6 medicina-58-00532-f006:**
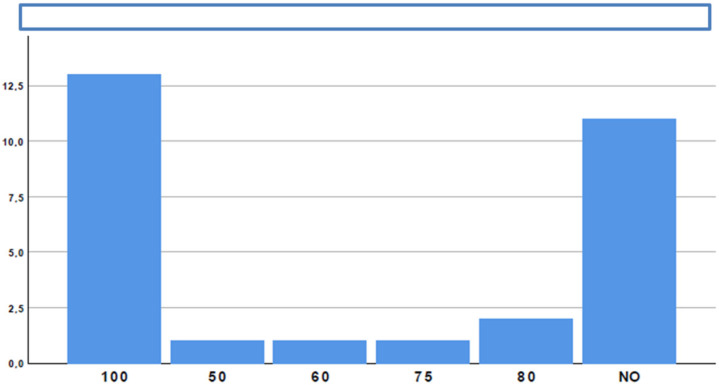
Perceived rate of application of EUS-GEA in the palliative setting.

**Figure 7 medicina-58-00532-f007:**
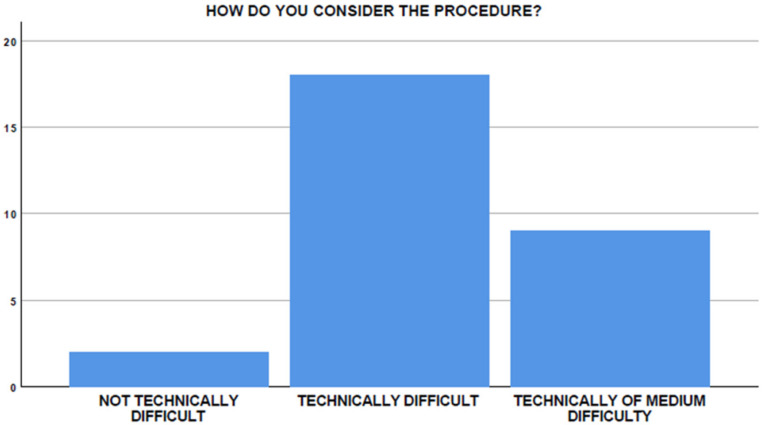
Perceived difficulty of EUS-GEA.

**Figure 8 medicina-58-00532-f008:**
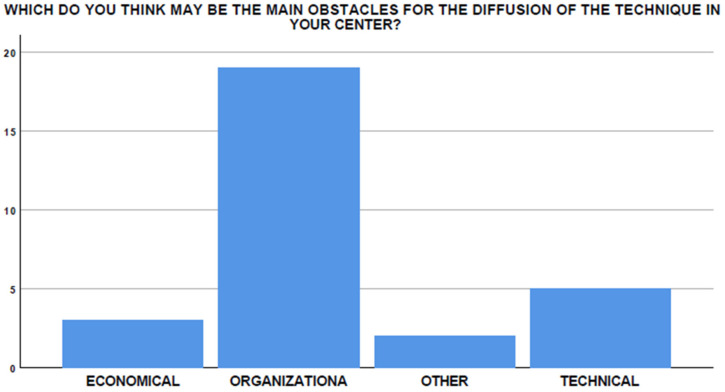
Perceived obstacles in EUS-GEA diffusion in the setting of afferent limb syndrome.

**Figure 9 medicina-58-00532-f009:**
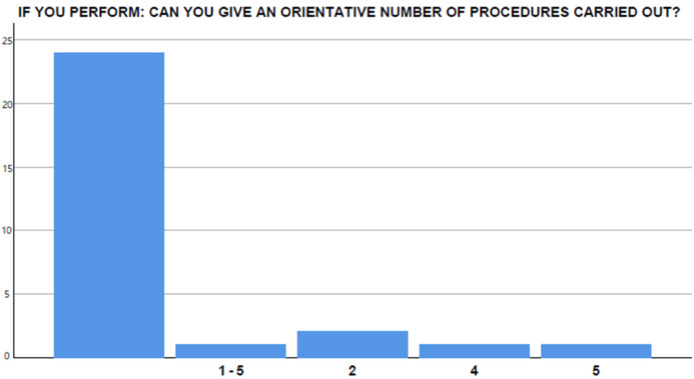
EUS-GG endoscopist’s experience.

**Figure 10 medicina-58-00532-f010:**
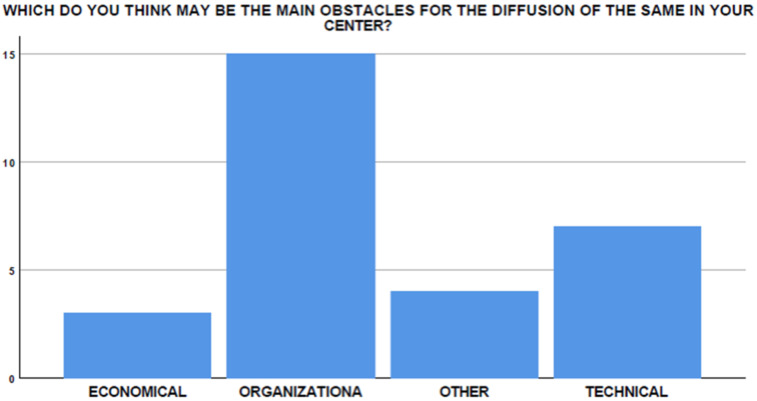
Perceived obstacles to diffusion of the technique in different centers.

**Figure 11 medicina-58-00532-f011:**
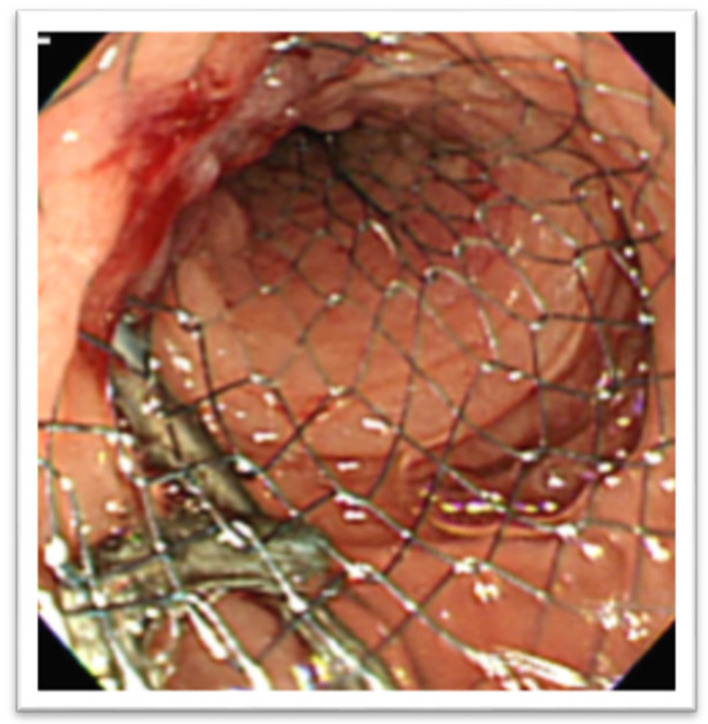
Duodenal self-expandable metal stent (SEMS).

**Figure 12 medicina-58-00532-f012:**
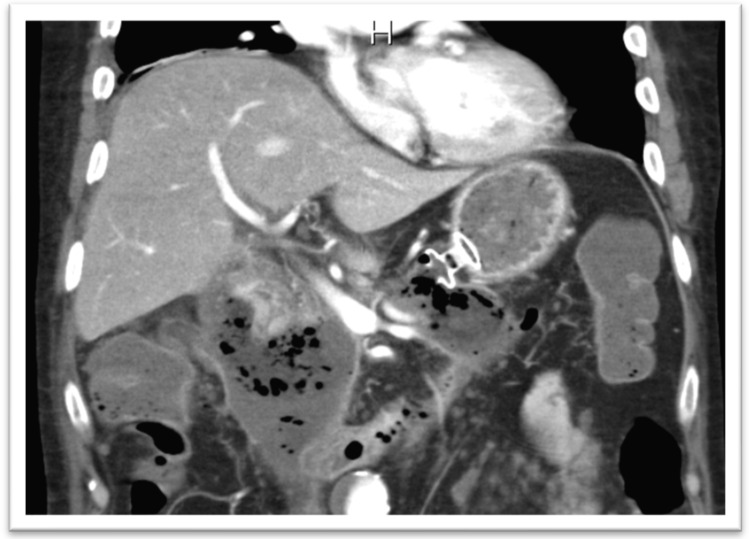
Lumen-apposing metal stent in the context of EUS-GEA.

## Data Availability

Not applicable.

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
