# Peer review of "Perceived Feasibility of Endoscopic Ultrasound-Guided Gastroenteric Anastomosis: An Italian Survey"

_medicina, 2022, doi:10.3390/medicina58040532_

Round 1

Reviewer 1 Report

The article has been revised well, and no further comments are necessary. 

Author Response

Dear reviewer, many thanks for Your valuable appreciation.

Sincerely

Reviewer 2 Report

I have no more comments, the changes are satisfactory

Author Response

Dear reviewer, many thanks for Your valuable appreciation.

Sincerely

This manuscript is a resubmission of an earlier submission. The following is a list of the peer review reports and author responses from that submission.

Round 1

Reviewer 1 Report

This is a very important study to assess the current experience with EUS guided procedures in Italy. This is first of its kind survey to understand clinical practice of EUS-GEA in Italy. the survey results are surprising but emphasizes the need for standardization of the indication, complication rates for the procedure, perhaps a national collaborative database might be helpful. 

Author Response

Response to reviewers’ comments

Dear Editors, dear Reviewers,

We wish to express our appreciation to the Editors and Reviewers for their insightful comments, which have helped us significantly to improve our manuscript. According to the suggestions, we have thoroughly revised our manuscript and its final version is enclosed. Point-by-point responses to the comments are listed below.

Reviewers’ comments #1

- This is a very important study to assess the current experience with EUS guided procedures in Italy. This is first of its kind survey to understand clinical practice of EUS-GEA in Italy. the survey results are surprising but emphasizes the need for standardization of the indication, complication rates for the procedure, perhaps a national collaborative database might be helpful. 

Response: Dear reviewer, many thanks for Your valuable comment and for Your appreciation.

Reviewer 2 Report

The authors reported the feasibility of EUS-guided gastro-enteric anastomosis. 

The field of EUS-guided drainages and anastomosis is emerging and indefinitely important, as this interesting procedure eventually will provide us with an improved capabilities to treat our patients. In fact, one of the main limitation of these procedures is the learning curve, as to date still there is no consensus regarding the feasibility among endosonographers.

Overall, this is a novel and interesting study that worth to be published.

I only have one minor comment: Make the headings with dark black color, while the rest of the text to be with normal black color without addressing the "B" function within the microsoft word file, and unify all the text and the words within the figures to be the same font and the same size.

Author Response

Reviewers’ comments #2

 - The field of EUS-guided drainages and anastomosis is emerging and indefinitely important, as this interesting procedure eventually will provide us with an improved capabilities to treat our patients. In fact, one of the main limitation of these procedures is the learning curve, as to date still there is no consensus regarding the feasibility among endosonographers.

Overall, this is a novel and interesting study that worth to be published.

I only have one minor comment: Make the headings with dark black color, while the rest of the text to be with normal black color without addressing the "B" function within the microsoft word file, and unify all the text and the words within the figures to be the same font and the same size.

Response: Dear reviewer, we corrected the style of the text

Reviewer 3 Report

This article conducted the prospective feasibility of endoscopic ultrasound-guided gastroenteric anastomosis with a web-based questionnaire in an Italian cohort. It is an interesting article; however, some concerns about this article. 1. There is no definition of skilled/experts of the endoscopist in the article. 2. The authors could present the figures of EUS-GEA, LAMS, SEMS, and for better understanding for general readers.3. The authors should also consider the patient QOL and prognosis instead of the high-endoscopic procedure techniques in the article. 4. The article may be acceptable; however, it needs more English text editing. 5. The number of references is small.

Author Response

Reviewers’ comments #3

  • This article conducted the prospective feasibility of endoscopic ultrasound-guided gastroenteric anastomosis with a web-based questionnaire in an Italian cohort. It is an interesting article; however, some concerns about this article. 1. There is no definition of skilled/experts of the endoscopist in the article. 2. The authors could present the figures of EUS-GEA, LAMS, SEMS, and for better understanding for general readers.3. The authors should also consider the patient QOL and prognosis instead of the high-endoscopic procedure techniques in the article. 4. The article may be acceptable; however, it needs more English text editing. The number of references is small.

Response: Dear reviewer, many thanks for Your valuable suggestions.

  1. We defined a skilled endosonographer in the introduction. The 3 experts were the founder of i-EUS group, with an experience in EUS lasting more than 20 years
  2. We presented in the text the figures of EUS-GEA, LAMS and SEMS
  3. We added in the discussion that, since our study was a survey about retrospective data, it was not possible to obtain findings about QoL and patients’ prognosis.
  4. We edited the text after a review by a native English speaker
  5. We added further references to the text, as highlighted in the revised version of the manuscript

Many thanks again

Sincerely Yours

Ilaria Tarantino & Emanuele Sinagra